# Three-Dimensional Non-Stationary MIMO Channel Modeling for UAV-Based Terahertz Wireless Communication Systems

**DOI:** 10.3390/e27080788

**Published:** 2025-07-25

**Authors:** Kai Zhang, Yongjun Li, Xiang Wang, Zhaohui Yang, Fenglei Zhang, Ke Wang, Zhe Zhao, Yun Wang

**Affiliations:** 1School of Information and Navigation, Air Force Engineering University, Xi’an 710082, China; 13262733921@163.com (K.Z.); wangke12345@stu.xjtu.edu.cn (K.W.); zhezhao@stu.xidian.edu.cn (Z.Z.); 2College of Information Science and Electronic Engineering, Zhejiang University, Hangzhou 310027, China; yang_zhaohui@zju.edu.cn; 3The 93135 Armed Force of China PLA, Beijing, China; wangyipingguo11@163.com

**Keywords:** multiple-input multiple-output (MIMO), unmanned aerial vehicle (UAV), terahertz (THz), air-to-air (A2A), channel modeling

## Abstract

Terahertz (THz) wireless communications can support ultra-high data rates and secure wireless links with miniaturized devices for unmanned aerial vehicle (UAV) communications. In this paper, a three-dimensional (3D) non-stationary geometry-based stochastic channel model (GSCM) is proposed for multiple-input multiple-output (MIMO) communication links between the UAVs in the THz band. The proposed channel model considers not only the 3D scattering and reflection scenarios (i.e., reflection and scattering fading) but also the atmospheric molecule absorption attenuation, arbitrary 3D trajectory, and antenna arrays of both terminals. In addition, the statistical properties of the proposed GSCM (i.e., the time auto-correlation function (T-ACF), space cross-correlation function (S-CCF), and Doppler power spectrum density (DPSD)) are derived and analyzed under several important UAV-related parameters and different carrier frequencies, including millimeter wave (mmWave) and THz bands. Finally, the good agreement between the simulated results and corresponding theoretical ones demonstrates the correctness of the proposed GSCM, and some useful observations are provided for the system design and performance evaluation of UAV-based air-to-air (A2A) THz-MIMO wireless communications.

## 1. Introduction

### 1.1. Motivation

Driven by the new application requirements, the sixth-generation mobility communication technology (6G) has to introduce new technical requirements and performance metrics [1]. The peak data for fifth-generation mobility communication technology (5G) networks are in the order of 20 GB/s, while for 6G networks, they are expected to reach 1–10 TB/s due to the use of terahertz (THz) and optical wireless bands. The user-experienced data rate can achieve gigabyte-per-second levels with the aid of ultra-high-frequency bands. To provide global coverage, 6G wireless communication networks are expected to expand from terrestrial communication networks in first-generation mobility communication technology (1G) to 5G to space-air-ground-sea integrated networks (SAGSINs), including satellites, unmanned aerial vehicles (UAVs), terrestrial ultra-dense networks (UDNs), maritime communications, underwater acoustic communications, and so on. SAGSINs and THz wireless communication technology, as the key technologies of 6G [1], have gained significant attention in recent years. In addition, the multiple-input multiple-output (MIMO) communication technology is often utilized to compensate for the transmission loss of wireless channels.

6G-enabling technologies aim to greatly increase the channel capacity, which is approximated through the summation of Shannon capacities of various types of wireless channels by considering interference in the wireless communication environment.

To realize 6G networks with these new enabling technologies and new performance metrics, the wireless channels of 6G-enabling technologies need to be thoroughly studied. In addition, the study of wireless channels is the foundation and essential for the design of wireless communication systems, network optimization, and performance evaluation in the context of 6G-enabling networks.

### 1.2. Related Work

#### 1.2.1. UAV-Based Channels

In SAGINs, UAVs are widely used in many scenarios, serving as airborne platforms for the wireless communication systems, i.e., relay station, base-station, and terminal-device in the air. UAVs have a number of crucial characteristics which make them optimal candidates for certain application scenarios, such as rapid three-dimensional (3D) deployment, arbitrary trajectory, low cost, and high mobility.

In general, UAVs have two typical application scenarios in wireless communications: air-to-ground (A2G) [2,3,4,5,6,7,8,9,10,11] and air-to-air (A2A) [12,13,14]. The usually applied frequency bands are the microwave and mmWave bands. However, to the best of our knowledge, there are few studies considering bands above 100 GHz for UAV-based wireless channels.

In the existing literature, the geometry-based stochastic model (GBSM) is often utilized to model the UAV-based wireless channels in different scenarios. As the wireless channels of UAV-based communications are time-varying, the wireless channel properties, e.g., the number of paths, average fading duration (AFD) [2,3,4,5,6,7], space–time correlation (STC) [6,7,8,9,10,11], path gain, Doppler power spectrum density (DPSD), and dynamic angle parameters (e.g., angle of departure (AoD), and angle of arrival (AoA)) [2,3,4,7,8,10,11], have been extensively investigated in different scenarios and carrier frequencies (e.g., microwave [2,3,4,5,6,7,8,9,13] and mmWave bands [10,11,12]).

The authors proposed a non-stationary wideband A2G MIMO channel model based on aeronautic random mobility model in the 2 GHz band, and the proposed channel model fully considers the random movement trajectory and high mobility of transmitter and receiver in [2]. In [12], the authors proposed a 3D non-stationary geometric channel model which consisted of the sum of the line-of-sight (LoS) and non-LoS (NLoS) propagation paths for the wideband MIMO A2A channels in the mmWave bands (such as 28 GHz and 60 GHz). For this proposed channel model, a two-state continuous-time Markov process is utilized to model the dynamic evolution (including birth, survival, and death) of scatterers at the propagation environment.

#### 1.2.2. THz Channel

In general, the terahertz range refers to the 0.1–10 THz band, while the millimeter wave (mmWave) denotes the 30–300 GHz band. Then, the 100–300 GHz band shares some common properties with mmWave and terahertz, e.g., ultra-large bandwidth, high directivity, large path propagation loss, blockage effects, atmospheric absorption attenuation, and more scattering and reflection fading on the rough surfaces [1]. The path loss, atmospheric absorption attenuation, and scattering and refection fading on the rough surfaces of the THz bands are more severe than those of mmWave bands [14,15,16].

The existing literature clearly shows that the study of wireless communication in the THz bands is still in its infancy. The study of THz wireless channels has been investigated through several common methods, such as measurement [17,18,19,20,21,22,23,24,25,26,27,28], channel modeling [25,29,30,31,32,33,34,35,36,37,38], and performance analysis in specific scenarios, such as indoor and outdoor.

So far, there are a lot of measurements for the wireless channel in THz band, and channel properties have been analyzed in many literature. Most current THz channel measurements are below the 300 GHz band. While the channel characteristics above 300 GHz are still few, not clear, and need extensive channel measurements in the future. In [27], the two different indoor scenario measurements were conducted at the wideband THz (i.e., 300 GHz bands). In [28], some preliminary path loss, partition loss, and scattering fading measurements were conducted at 100 GHz.

For the THz channel model, the deterministic model and stochastic model are often utilized to channel model of the THz wireless channel in different scenarios. In general, the THz channel models consisting of the sum of the LoS and NLoS propagation paths, where the NLoS path includes the reflection, scattering, and diffracted propagation paths. The deterministic channel model includes a ray-tracing channel model, map-based channel model, and point cloud channel model. Moreover, the stochastic channel model includes GBSM, a correlation-based stochastic model (CBSM), and a beam domain stochastic model (BDSM) [1].

In addition, the statistical properties of the wireless channels have been investigated, including the path loss, propagation fading on the rough surfaces [25], and space–time–frequency correlation (STFC) [25,34,35,36,37,38]. In [37], the authors propose a THz channel model for indoor scenario based on the ray-tracing method. The statistical properties of different paths, including LoS, reflection, scattering, and diffraction paths, have been studied (in Figure 1) [30]. In [38], the authors propose a 3D hybrid dynamic channel model for indoor THz communications. In [25], the authors propose a general 3D space–time–frequency non-stationary THz channel model for ultra-massive wireless communication systems, and the channel statistical properties of the proposed channel model are analyzed in space, time, and frequency domains.

#### 1.2.3. UAV-Based THz Channel

In addition, a few studies have been conducted on the channel modeling of UAV-based wireless communication in sub-THz and THz bands [39,40,41,42,43]. In the existing literature, the 3D time-varying channel models of UAV-based wireless communication have been investigated in A2G [39,41,43] and A2A [40,42,43] scenarios, and the statistical properties of the channels, e.g., path loss, channel capacity, STC, and DPSD, have been analyzed. In [39,40], the authors propose 3D geometry-based stochastic channel models for the A2G and A2A UAV-based MIMO wireless communications, respectively.

However, they only consider the influence of reflection and scattering fading on the rough surface on wireless propagation and do not consider the influence of atmospheric absorption attenuation on wireless propagation in [39,40]. In addition, the authors consider the influence of reflection and scattering fading on the rough surfaces and atmospheric absorption attenuation on wireless propagation for the A2A and A2G UAV-based SISO wireless communications. However, the channel model and characteristics of UAV-based MIMO wireless communications at the THz band are still not clear and need extensive channel model, measurements, and performance analysis in the future.

### 1.3. Contribution of Paper

In this paper, we present a novel time-varying GSCM of MIMO wireless channels for the THz UAV-based communications. This proposed GSCM accurately characterizes the time-varying UAV-MIMO wireless channel scenario in the THz band, including the reflection and scattering fading, the atmospheric absorption attenuation, and 3D arbitrary trajectories and antenna arrays of both terminals. In addition, some theoretical statistical properties of the proposed GSCM, i.e., the T-ACF, the S-CCF, and the DPSD, are analyzed and derived.

The remainder of this paper is organized as follows: Section 2 proposes a 3D non-stationary THz UAV-based MIMO channel model. The statistical properties of the proposed THz UAV-based MIMO channel model are derived in Section 3. Some numerical results and analysis are given in Section 4. Finally, the conclusions are drawn in Section 5.

## 2. 3D Non-Stationary UAV-MIMO Channel Model

### 2.1. Description of the THz UAV-MIMO Communication System

In this paper, we assume that the antennas at the transmitter (Tx) and receiver (Rx) are both isotropic [9] and antenna arrays for the UAV-based THz-MIMO communication system. The antennas at Tx and Rx are both located on the UAVs, and the number of antenna at Tx and Rx are *P* and *Q*, respectively. The UAV-based communication has some characteristics, i.e., 3D random deployment and 3D arbitrary trajectory, flexible environment adaptability, and high space–time variation. The typical UAV-based A2A wireless communication scenario in the THz band is shown in Figure 2, where the propagation paths between UAVs are time–space variant. For the time–space variant channel, the multipaths are informed in wireless environments with free space, obstructions, and rough surfaces. Therefore, the propagation paths can be categorized into LoS and NLoS cases, as shown in Figure 3. Due to the severe propagation path loss of the wireless communication link between the UAVs in THz bands, i.e., spread loss, scattering and reflection fading, and atmospheric absorption attenuation, the NLoS path of UAV-based communication considers only a single bounce in the communication scenario of this paper.

In Figure 3, the coordinates of the *p*-th (*p* = 1, 2, …, *P*) antenna element of antenna array at Tx and *q*-th (*q* = 1, 2, …, *Q*) antenna element of antenna array at Rx are denoted as $\left(x_{p}\left(t\right),y_{p}\left(t\right),z_{p}\left(t\right)\right)$ and $\left(x_{q}\left(t\right),y_{q}\left(t\right),z_{q}\left(t\right)\right)$, respectively. In Figure 3, *O* is the original of the coordinate system. The antenna element spacing of the antenna array at Tx and Rx are denoted as $d_{T}$ and $d_{R}$, respectively.

Here, we consider that the height of Tx’s UAV is higher than that of Rx, and the scattering and reflection fading on the rough surfaces only happen at Rx. In addition, in Figure 3b, $S_{m}$ denotes the *m*-th scatterer around the Rx (where *m* = 1, 2, …, *M*, *M* is the number of efficient scatterers), and $R_{n}$ denotes the *n*-th reflector around the Rx (where *n* = 1, 2, …, *N*, *N* is the number of efficient reflectors). As the UAVs at Tx and Rx are always moving, the coordinates of reflector $R_{n}$ and scatterer $S_{m}$ are denoted as $\left(x_{n}\left(t\right),y_{n}\left(t\right),z_{n}\left(t\right)\right)$ and $\left(x_{m}\left(t\right),y_{m}\left(t\right),z_{m}\left(t\right)\right)$, respectively. The other parameters of the UAV-based THz communication system are shown in Table 1.

According to the transmitting theory of wireless signals [7], the received signal $r\left(t\right)={\left[r_{1}\left(t\right),r_{2}\left(t\right),\ldots ,r_{Q}\left(t\right)\right]}^{T}$ of a MIMO wireless communication system is a function of the channel impulse response (CIR) matrix $H\left(t\right)={\left[h_{qp}\left(t\right)\right]}_{Q\times P}$ (where q=1,…,Q; p=1,…,P), the transmitted signal $s\left(t\right)={\left[s_{1}\left(t\right),s_{2}\left(t\right),\ldots ,s_{P}\left(t\right)\right]}^{T}$, and the wireless channel noise $n\left(t\right)=[n_{1}\left(t\right),n_{2}\left(t\right),\ldots ,n_{Q}\left(t\right)]$, which can be expressed as r(t)=H(t)∗s(t)+n(t).

In the time-varying wireless communications system of this paper, the CIR of *l*-th (where l=1,2,…,L(t)), propagation path between the *p*-th (where p=1,2,…,P), antenna at Tx and *q*-th (where q=1,2,…,Q), and antenna at Rx can be further calculated as functions of the time-varying channel gain value $\alpha_{l,qp}\left(t\right)$, time-varying propagation path delay $\tau_{l,qp}\left(t\right)$, carrier frequency $f_{c}$, and time-varying Doppler frequency shift $f_{D,l,qp}\left(t\right)$, which can be expressed as follows:(1)$$\begin{array}{c} \begin{array}{c} h_{qp}\left(t\right)=\sum_{l=1}^{L(t)}\alpha_{l,qp}\left(t\right)e^{j[2\pi tf_{D,l,qp}\left(t\right)-2\pi f_{c}\tau_{l,qp}\left(t\right)]} \end{array} \end{array}$$
where L(t) is the number of the time-varying multipath components (MPCs). In this paper, we consider the LoS, reflection, and scattering paths in the channel model to be independent each other. Then, Equation (Equation 1) can be expressed as follows:(2)$$\begin{array}{c} h_{qp}\left(t\right)=h_{qp}^{\mathrm{LoS}}\left(t\right)+h_{qp}^{\mathrm{ref}}\left(t\right)+h_{qp}^{\mathrm{sca}}\left(t\right) \end{array}$$

In general, the heights of Tx and Rx for the UAV-based A2A communication system are always time-varying, which is caused by the mobility of the UAVs of Tx and Rx. According to the communication scenario in Figure 2, the probability of scattering and reflection propagation fading around the Rx are higher than those around the Tx. Additionally, the radio-wave wavelength of the THz band is much smaller than that of the mmWave and microwave bands. Therefore, the probability of scattering and reflection propagations on the rough surfaces with different materials in THz band is larger than that in mmWave and microwave bands. In addition, the influence of the atmospheric molecule absorption attenuation $A_{\mathrm{ma}}\left(f_{c},t\right)$ on the wireless channel increases rapidly with the increasing of the radio-wave frequency. In particular, $A_{\mathrm{ma}}\left(f_{c},t\right)$ in the THz band is higher than that in mmWave and microwave bands [44].

### 2.2. CIRs of the UAV-Based A2A Wireless Channel

This paper categorizes propagation paths into two types: LoS and NLoS (including reflection and scattering propagation paths) scenarios, as shown in Figure 3a,b.

#### 2.2.1. LoS Propagation Path

In a time-varying LoS propagation channel of this paper, there are no obstructions between the propagation paths from Tx to Rx, as shown in Figure 3a. Then, the CIR of the time-varying LoS propagation path between the *p*-th antenna element of the transmitting antenna array and the *q*-th antenna element of the receiving antenna array can be written as follows:(3)$$\begin{array}{c} \begin{array}{c} h_{qp}^{\mathrm{LoS}}\left(t\right)=\sqrt{\frac{\Omega_{qp}\left(t\right)K_{qp}\left(t\right)}{K_{qp}\left(t\right)+1}}\;\cdot \;e^{j\{2\pi t\left[f_{D,\mathrm{LoS},p}^{\mathrm{Tx}}\left(t\right)+f_{D,\mathrm{LoS},q}^{\mathrm{Rx}}\left(t\right)\right]-\frac{2\pi d_{\mathrm{LoS},qp}\left(t\right)}{\lambda_{c}}\}} \end{array} \end{array}$$
where $d_{\mathrm{LoS},qp}\left(t\right)$ is the 3D distance between the *p*-th antenna element of the transmitting antenna array and the *q*-th antenna element of the receiving antenna array for the LoS propagation path; $\lambda_{c}=c/f_{c}$ is the wavelength of radio-wave; *c* is the speed of light; $K_{qp}\left(t\right)$ is the Rician factor of the wireless propagation link between the *p*-th antenna element of the transmitting antenna array and the *q*-th antenna element of the receiving antenna array, and we assumed that $K_{qp}\left(t\right)=K_{qp}$ in this paper. $\Omega_{qp}\left(t\right)=E[|h_{qp}\left(t\right){|}^{2}]$ (where E[·] is the statistical expectation operation) is the propagation path power from the *p*-th antenna element of the transmitting antenna array and the *q*-th antenna element of the receiving antenna array.

In Equation (Equation 3), $f_{D,\mathrm{LoS},p}^{\mathrm{Tx}}\left(t\right)$ and $f_{D,\mathrm{LoS},q}^{\mathrm{Rx}}\left(t\right)$ denote the Doppler frequency shift at the *p*-th antenna element of the transmitting antenna array and the *q*-th antenna element of the receiving antenna array respectively, which can be expressed as follows:(4)$$\begin{array}{c} \begin{array}{c} f_{D,\mathrm{LoS},p}^{\mathrm{Tx}}\left(t\right)=\frac{v_{\mathrm{Tx}}}{\lambda_{c}}\left[\cos\left(\theta_{\mathrm{LoS},p}^{\mathrm{AoD}}\left(t\right)-\theta_{\mathrm{Tx}}\right)\;\cdot \;\cos\left(\phi_{\mathrm{LoS},p}^{\mathrm{AoD}}\left(t\right)\right)\cos\left(\phi_{\mathrm{Tx}}\right)+\sin\left(\phi_{\mathrm{LoS},p}^{\mathrm{AoD}}\left(t\right)\right)\sin\left(\phi_{\mathrm{Tx}}\right)\right] \end{array} \end{array}$$(5)$$\begin{array}{c} \begin{array}{c} f_{D,\mathrm{LoS},q}^{\mathrm{Rx}}\left(t\right)=\frac{v_{\mathrm{Rx}}}{\lambda_{c}}\left[\cos\left(\theta_{\mathrm{LoS},q}^{\mathrm{AoA}}\left(t\right)-\theta_{\mathrm{Rx}}\right)\;\cdot \;\cos\left(\phi_{\mathrm{LoS},q}^{\mathrm{AoA}}\left(t\right)\right)\cos\left(\phi_{\mathrm{Rx}}\right)+\sin\left(\phi_{\mathrm{LoS},q}^{\mathrm{AoA}}\left(t\right)\right)\sin\left(\phi_{\mathrm{Rx}}\right)\right] \end{array} \end{array}$$

#### 2.2.2. NLoS Propagation Path

For the NLoS propagation path of this paper, it includes both the reflection and scattering propagation paths, as shown in Figure 3b. According to the described above, the NLoS path considers only a single bounce in this paper. Moreover, the efficient points’ locations of the reflection and scattering propagation near the Rx are assumed to be uniformly and randomly distributed in this paper.

Then, given the Cartesian coordinate of Tx, Rx, and the points of scattering and reflection, the propagation distance of each reflected and scattered ray from the *p*-th antenna element of the transmitted antenna array to the *q*-th antenna element of the received antenna array at the *n*-th reflector and *m*-th scatterer are calculated, respectively, as follows: $d_{n,qp}^{\mathrm{ref}}\left(t\right)=|d_{\mathrm{Tx},np}^{\mathrm{ref}}\left(t\right)|+|d_{\mathrm{Rx},qn}^{\mathrm{ref}}\left(t\right)|$ and $d_{m,qp}^{\mathrm{sca}}\left(t\right)=|d_{\mathrm{Tx},mp}^{\mathrm{sca}}\left(t\right)|+|d_{\mathrm{Rx},qm}^{\mathrm{sca}}\left(t\right)|$, where $d_{\mathrm{Tx},np}^{\mathrm{ref}}\left(t\right)$ and $d_{\mathrm{Rx},qn}^{\mathrm{ref}}\left(t\right)$ denote the propagation distances of the *n*-th reflector, *p*-th antenna element of the transmitted antenna array, and *q*-th antenna element of the received antenna array for the reflection propagation path, respectively; $d_{\mathrm{Tx},mp}^{\mathrm{sca}}\left(t\right)$ and $d_{\mathrm{Rx},qm}^{\mathrm{sca}}\left(t\right)$ denote the propagation distances by the *m*-th scatterer and *p*-th antenna element of the transmitted antenna array and *q*-th antenna element of received antenna array for the scattering propagation path, respectively.

According to the above-described, the CIRs of the time-varying reflection and scattering propagation paths between the *p*-th antenna element of the transmitting antenna array and the *q*-th antenna element of the receiving antenna array can be written as follows:(6)$$\begin{array}{c} \begin{array}{c} h_{qp}^{\mathrm{ref}}\left(t\right)=\sqrt{\frac{\Omega_{qp}\left(t\right)\eta_{\mathrm{ref},qp}\left(t\right)}{N[K_{qp}\left(t\right)+1]}}\sum_{n=1}^{N}\alpha_{n,qp}^{\mathrm{ref}}\left(t\right)e^{j\vartheta_{\mathrm{ref},n}}\;\cdot \;e^{j\{2\pi t\left[f_{D,np}^{\mathrm{Tx},\mathrm{ref}}\left(t\right)+f_{D,qn}^{\mathrm{Rx},\mathrm{ref}}\left(t\right)\right]-\frac{2\pi d_{n,qp}^{\mathrm{ref}}\left(t\right)}{\lambda_{c}}\}} \end{array} \end{array}$$(7)$$\begin{array}{c} \begin{array}{c} h_{qp}^{\mathrm{sca}}\left(t\right)=\sqrt{\frac{\Omega_{qp}\left(t\right)\eta_{\mathrm{sca},qp}\left(t\right)}{M[K_{qp}\left(t\right)+1]}}\sum_{m=1}^{M}\alpha_{m,qp}^{\mathrm{sca}}\left(t\right)e^{j\vartheta_{\mathrm{sca},m}}\;\cdot \;e^{j\{2\pi t\left[f_{D,mp}^{\mathrm{Tx},\mathrm{sca}}\left(t\right)+f_{D,qm}^{\mathrm{Rx},\mathrm{sca}}\left(t\right)\right]-\frac{2\pi d_{m,qp}^{\mathrm{sca}}\left(t\right)}{\lambda_{c}}\}} \end{array} \end{array}$$
with the Doppler frequency shift terms $f_{D,np}^{\mathrm{Tx},\mathrm{ref}}\left(t\right)$, $f_{D,qn}^{\mathrm{Rx},\mathrm{ref}}\left(t\right)$, $f_{D,mp}^{\mathrm{Tx},\mathrm{sca}}\left(t\right)$, and $f_{D,qm}^{\mathrm{Rx},\mathrm{sca}}\left(t\right)$, which can be expressed as follows:(8)$$\begin{array}{c} \begin{array}{c} f_{D,np}^{\mathrm{Tx},\mathrm{ref}}\left(t\right)=\frac{v_{\mathrm{Tx}}}{\lambda_{c}}\left[\cos\left(\theta_{\mathrm{ref},np}^{\mathrm{AoD}}\left(t\right)-\theta_{\mathrm{Tx}}\right)\;\cdot \;\cos\left(\phi_{\mathrm{ref},np}^{\mathrm{AoD}}\left(t\right)\right)\;\cdot \;\cos\left(\phi_{\mathrm{Tx}}\right)+\sin\left(\phi_{\mathrm{ref},np}^{\mathrm{AoD}}\left(t\right)\right)\sin\left(\phi_{\mathrm{Tx}}\right)\right] \end{array} \end{array}$$(9)$$\begin{array}{c} \begin{array}{c} f_{D,qn}^{\mathrm{Rx},\mathrm{ref}}\left(t\right)=\frac{v_{\mathrm{Rx}}}{\lambda_{c}}\left[\cos\left(\theta_{\mathrm{ref},qn}^{\mathrm{AoA}}\left(t\right)-\theta_{\mathrm{Rx}}\right)\;\cdot \;\cos\left(\phi_{\mathrm{ref},qn}^{\mathrm{AoA}}\left(t\right)\right)\;\cdot \;\cos\left(\phi_{\mathrm{Rx}}\right)+\sin\left(\phi_{\mathrm{ref},qn}^{\mathrm{AoA}}\left(t\right)\right)\sin\left(\phi_{\mathrm{Rx}}\right)\right] \end{array} \end{array}$$(10)$$\begin{array}{c} \begin{array}{c} f_{D,mp}^{\mathrm{Tx},\mathrm{sca}}\left(t\right)=\frac{v_{\mathrm{Tx}}}{\lambda_{c}}\left[\cos\left(\theta_{\mathrm{sca},mp}^{\mathrm{AoD}}\left(t\right)-\theta_{\mathrm{Tx}}\right)\;\cdot \;\cos\left(\phi_{\mathrm{sca},mp}^{\mathrm{AoD}}\left(t\right)\right)\;\cdot \;\cos\left(\phi_{\mathrm{Tx}}\right)+\sin\left(\phi_{\mathrm{sca},mp}^{\mathrm{AoD}}\left(t\right)\right)\sin\left(\phi_{\mathrm{Tx}}\right)\right] \end{array} \end{array}$$(11)$$\begin{array}{c} \begin{array}{c} f_{D,qm}^{\mathrm{Rx},\mathrm{sca}}\left(t\right)=\frac{v_{\mathrm{Rx}}}{\lambda_{c}}\left[\cos\left(\theta_{\mathrm{sca},qm}^{\mathrm{AoA}}\left(t\right)-\theta_{\mathrm{Rx}}\right)\;\cdot \;\cos\left(\phi_{\mathrm{sca},qm}^{\mathrm{AoA}}\left(t\right)\right)\;\cdot \;\cos\left(\phi_{\mathrm{Rx}}\right)+\sin\left(\phi_{\mathrm{sca},qm}^{\mathrm{AoA}}\left(t\right)\right)\sin\left(\phi_{\mathrm{Rx}}\right)\right] \end{array} \end{array}$$

According to the attenuation experience model of oxygen and water vapor molecules in ITU-R P.676-11 [44], the Kirchhoff scattering theory [45], the modified Beckmann–Kirchhoff theory [45,46], and the Friis formular, the channel coefficients of the reflection and the scattering components can be calculated as follows:(12)$$\begin{array}{c} \begin{array}{cc} \; & \alpha_{n,qp}^{\mathrm{ref}}\left(t\right)=\frac{\frac{\lambda_{c}^{2}\;\cdot \;R_{n,qp}\left(f_{c},t\right)\;\cdot \;a_{\mathrm{ma},n,qp}^{\mathrm{ref}}\left(f_{c},t\right)}{16\pi^{2}d_{\mathrm{Tx},np}^{\mathrm{ref}}\left(t\right)\;\cdot \;d_{\mathrm{Rx},qn}^{\mathrm{ref}}\left(t\right)}}{\sqrt{\frac{1}{N}\sum_{n}^{N}{\left[\frac{\lambda_{c}^{2}\;\cdot \;R_{n,qp}\left(f_{c},t\right)\;\cdot \;a_{\mathrm{ma},n,qp}^{\mathrm{ref}}\left(f_{c},t\right)}{16\pi^{2}d_{\mathrm{Tx},np}^{\mathrm{ref}}\left(t\right)\;\cdot \;d_{\mathrm{Rx},qn}^{\mathrm{ref}}\left(t\right)}\right]}^{2}}} \end{array} \end{array}$$(13)$$\begin{array}{c} \begin{array}{cc} \; & \alpha_{m,qp}^{\mathrm{sca}}\left(t\right)=\frac{\frac{\lambda_{c}^{2}\;\cdot \;S_{m,qp}\left(f_{c},t\right)\;\cdot \;a_{\mathrm{ma},m,qp}^{\mathrm{sca}}\left(f_{c},t\right)}{16\pi^{2}d_{\mathrm{Tx},mp}^{\mathrm{sca}}\left(t\right)\;\cdot \;d_{\mathrm{Rx},qm}^{\mathrm{sca}}\left(t\right)}}{\sqrt{\frac{1}{M}\sum_{m}^{M}{\left[\frac{\lambda_{c}^{2}\;\cdot \;S_{m,qp}\left(f_{c},t\right)\;\cdot \;a_{\mathrm{ma},m,qp}^{\mathrm{sca}}\left(f_{c},t\right)}{16\pi^{2}d_{\mathrm{Tx},mp}^{\mathrm{sca}}\left(t\right)\;\cdot \;d_{\mathrm{Rx},qm}^{\mathrm{sca}}\left(t\right)}\right]}^{2}}} \end{array} \end{array}$$
where $R_{n,qp}\left(f_{c},t\right)$ and $S_{m,qp}\left(f_{m},t\right)$ are the coefficients of the reflection and scattering propragation paths on the rough surfaces for the link from the *p*-th antenna element of the transmitting antenna array to the *q*-th antenna element of the receiving antenna array, $a_{\mathrm{ma},n,qp}^{\mathrm{ref}}\left(f_{c},t\right)$ and $a_{\mathrm{ma},m,qp}^{\mathrm{sca}}\left(f_{c},t\right)$ are the channel gain caused by the atmospheric absorption in the reflection and scattering propagation paths, which can be calculated as:(14)$$\begin{array}{c} \begin{array}{cc} \; & a_{\mathrm{ma},n,qp}^{\mathrm{ref}}\left(f_{c},t\right)=e^{-\frac{1}{2}\left[A_{\mathrm{ma},np}^{\mathrm{ref}}\left(f_{c},t\right)\right]}\;\cdot \;e^{-\frac{1}{2}\left[A_{\mathrm{ma},qn}^{\mathrm{ref}}\left(f_{c},t\right)\right]} \end{array} \end{array}$$(15)$$\begin{array}{c} \begin{array}{cc} \; & a_{\mathrm{ma},m,qp}^{\mathrm{sca}}\left(f_{c},t\right)=e^{-\frac{1}{2}\left[A_{\mathrm{ma},mp}^{\mathrm{sca}}\left(f_{c},t\right)\right]}\;\cdot \;e^{-\frac{1}{2}\left[A_{\mathrm{ma},qm}^{\mathrm{sca}}\left(f_{c},t\right)\right]} \end{array} \end{array}$$

In Equations (14) and (15), $A_{\mathrm{ma},np}^{\mathrm{ref}}\left(f_{c},t\right)$ and $A_{\mathrm{ma},qn}^{\mathrm{ref}}\left(f_{c},t\right)$ denote the atmospheric absorption attenuation of the propagation link between the *p*-th antenna element of the transmitting antenna array and the *n*-th reflector and the atmospheric absorption attenuation of the propagation link between the *n*-th reflector and the *q*-th antenna element of the receiving antenna array, respectively; $A_{\mathrm{ma},mp}^{\mathrm{sca}}\left(f_{c},t\right)$ and $A_{\mathrm{ma},qm}^{\mathrm{sca}}\left(f_{c},t\right)$ denote the atmospheric absorption attenuation of the propagation link between the *p*-th antenna element of the transmitting antenna array and the *m*-th scatterer and the atmospheric absorption attenuation of the propagation link between the *m*-th scatterer and the *q*-th antenna element of the receiving antenna array, respectively.

## 3. Statistical Properties of the Proposed UAV-Based Air-to-Air MIMO Channel Model

In this section, some typical channel statistical properties for UAV-based A2A MIMO channels in the THz band will be derived, including the time–space correlation function (STCF) (i.e., time autocorrelation function (T-ACF) and space cross-correlation function (S-CCF)) and DPSD.

### 3.1. Space–Time Correlation Function

Since the time-varying CIRs of reflection, scattering, and LoS components are independent of each other and the complex processes in Equations (3), (6) and (7), the normalized STCF between two complex envelopes channels $h_{qp}\left(t\right)$ and $h_{q^{\prime }p^{\prime }}\left(t\right)$ can be calculated as follows:(16)$$\begin{array}{c} \begin{array}{cc} R_{qp,q^{\prime }p^{\prime }}\left(d_{T},d_{R},t,\Delta t\right) & =\frac{E[h_{qp}\left(t\right){\left(h_{q^{\prime }p^{\prime }}\left(t+\Delta t\right)\right)}^{*}]}{\sqrt{E[|\left(h_{qp}\left(t\right){|}^{2}\right]E[|{\left(h_{q^{\prime }p^{\prime }}\left(t+\Delta t\right)\right)}^{*}{|}^{2}]}}\\ \; & =\frac{E[h_{qp}\left(t\right){\left(h_{q^{\prime }p^{\prime }}\left(t+\Delta t\right)\right)}^{*}]}{\sqrt{\Omega_{qp}\left(t\right)\;\cdot \;\Omega_{q^{\prime }p^{\prime }}\left(t+\Delta t\right)}} \end{array} \end{array}$$
where E[·] and [·]* respectively denote the statistical expectation operator and complex conjugate operation.

According to the Equation (Equation 2), the Equation (Equation 16) can be simplified with the following:(17)$$\begin{array}{c} \begin{array}{c} R_{qp,q^{\prime }p^{\prime }}\left(d_{T},d_{R},t,\Delta t\right)=R_{qp,q^{\prime }p^{\prime }}^{\mathrm{LoS}}\left(d_{T},d_{R},t,\Delta t\right)+R_{qp,q^{\prime }p^{\prime }}^{,\mathrm{ref}}\left(d_{T},d_{R},t,\Delta t\right)+R_{qp,q^{\prime }p^{\prime }}^{\mathrm{sca}}\left(d_{T},d_{R},t,\Delta t\right) \end{array} \end{array}$$
where $R_{qp,q^{\prime }p^{\prime }}^{\mathrm{LoS}}\left(d_{T},d_{R},t,\Delta t\right)$, $R_{qp,q^{\prime }p^{\prime }}^{\mathrm{ref}}\left(d_{T},d_{R},t,\Delta t\right)$, and $R_{qp,q^{\prime }p^{\prime }}^{\mathrm{sca}}\left(d_{T},d_{R},t,\Delta t\right)$ denote the STCF for the LoS, reflection, and scattering components, which can be written as follows:(18)$$\begin{array}{c} \begin{array}{cc} R_{qp,q^{\prime }p^{\prime }}^{\mathrm{LoS}}\left(d_{T},d_{R},t,\Delta t\right) & =\frac{E[h_{\mathrm{LoS},qp}\left(t\right)\;\cdot \;{\left(h_{\mathrm{LoS},q^{\prime }p^{\prime }}\left(t+\Delta t\right)\right)}^{*}]}{\sqrt{\Omega_{qp}\left(t\right)\;\cdot \;\Omega_{q^{\prime }p^{\prime }}\left(t+\Delta t\right)}}\\ \; & =\sqrt{\frac{K_{qp}K_{q^{\prime }p^{\prime }}}{\left(K_{qp}+1\right)\left(K_{q^{\prime }p^{\prime }}+1\right)}}\;\cdot \;e^{-j\frac{2\pi }{\lambda_{c}}\left[d_{\mathrm{LoS},qp}\left(t\right)-d_{\mathrm{LoS},q^{\prime }p^{\prime }}\left(t+\Delta t\right)\right]}\\ \; & \times e^{j2\pi t[f_{D,\mathrm{LoS},p}^{\mathrm{Tx}}\left(t\right)+f_{D,\mathrm{LoS},q}^{\mathrm{Rx}}\left(t\right)]}\;\cdot \;e^{-j2\pi \left(t+\Delta t\right)[f_{D,\mathrm{LoS},p^{\prime }}^{\mathrm{Tx}}\left(t+\Delta t\right)+f_{D,\mathrm{LoS},q^{\prime }}^{\mathrm{Rx}}\left(t+\Delta t\right)]} \end{array} \end{array}$$(19)$$\begin{array}{c} \begin{array}{cc} R_{qp,q^{\prime }p^{\prime }}^{\mathrm{ref}}\left(d_{T},d_{R},t,\Delta t\right) & =\frac{E[h_{qp}^{\mathrm{ref}}\left(t\right)\;\cdot \;{\left(h_{q^{\prime }p^{\prime }}^{\mathrm{ref}}\left(t+\Delta t\right)\right)}^{*}]}{\sqrt{\Omega_{qp}\left(t\right)\;\cdot \;\Omega_{q^{\prime }p^{\prime }}\left(t+\Delta t\right)}}\\ \; & =\sqrt{\frac{\eta_{\mathrm{ref},qp}\eta_{\mathrm{ref},q^{\prime }p^{\prime }}}{\left(K_{qp}+1\right)\left(K_{q^{\prime }p^{\prime }}+1\right)}}\;\cdot \;\sum_{n=1}^{N}\alpha_{n,qp}^{\mathrm{ref}}\left(t\right)\;\cdot \;\alpha_{n,q^{\prime }p^{\prime }}^{\mathrm{ref}}\left(t+\Delta t\right)\\ \; & \times e^{-j\frac{2\pi }{\lambda_{c}}\left[d_{n,qp}^{\mathrm{ref}}\left(t\right)-d_{n,q^{\prime }p^{\prime }}^{\mathrm{ref}}\left(t+\Delta t\right)\right]}\;\cdot \;e^{j2\pi t[f_{D,np}^{\mathrm{Tx},\mathrm{ref}}\left(t\right)+f_{D,qn}^{\mathrm{Rx},\mathrm{ref}}\left(t\right)]}\\ \; & \times e^{-j2\pi \left(t+\Delta t\right)[f_{D,np^{\prime }}^{\mathrm{Tx},\mathrm{ref}}\left(t+\Delta t\right)+f_{D,q^{\prime }n}^{\mathrm{Rx},\mathrm{ref}}\left(t+\Delta t\right)]} \end{array} \end{array}$$(20)$$\begin{array}{c} \begin{array}{cc} R_{qp,q^{\prime }p^{\prime }}^{,\mathrm{sca}}\left(d_{T},d_{R},t,\Delta t\right) & =\frac{E[h_{qp}^{\mathrm{sca}}\left(t\right)\;\cdot \;{\left(h_{q^{\prime }p^{\prime }}^{\mathrm{sca}}\left(t+\Delta t\right)\right)}^{*}]}{\sqrt{\Omega_{qp}\left(t\right)\;\cdot \;\Omega_{q^{\prime }p^{\prime }}\left(t+\Delta t\right)}}\\ \; & =\sqrt{\frac{\eta_{\mathrm{sca},qp}\eta_{\mathrm{sca},q^{\prime }p^{\prime }}}{\left(K_{qp}+1\right)\left(K_{q^{\prime }p^{\prime }}+1\right)}}\;\cdot \;\sum_{m=1}^{M}\alpha_{m,qp}^{\mathrm{sca}}\left(t\right)\;\cdot \;\alpha_{m,q^{\prime }p^{\prime }}^{\mathrm{sca}}\left(t+\Delta t\right)\\ \; & \times e^{-j\frac{2\pi }{\lambda_{c}}\left[d_{m,qp}^{\mathrm{sca}}\left(t\right)-d_{m,q^{\prime }p^{\prime }}^{\mathrm{sca}}\left(t+\Delta t\right)\right]}\;\cdot \;e^{j2\pi t[f_{D,mp}^{\mathrm{Tx},\mathrm{sca}}\left(t\right)+f_{D,qm}^{\mathrm{Rx},\mathrm{sca}}\left(t\right)]}\\ \; & \times e^{-j2\pi \left(t+\Delta t\right)[f_{D,mp^{\prime }}^{\mathrm{Tx},\mathrm{sca}}\left(t+\Delta t\right)+f_{D,q^{\prime }m}^{\mathrm{Rx},\mathrm{sca}}\left(t+\Delta t\right)]} \end{array} \end{array}$$

In Equations (16)–(20), $d_{T}$ and $d_{R}$ denote the antenna element spacings of the ttransmitting antenna array and the receiving antenna array, respectively.

In this paper, the T-ACF and S-CCF can be obtained by setting $d_{T}=0$ and $d_{R}=0$, or Δt=0, which can be expressed as follows:(21)$$\begin{array}{c} \begin{array}{c} R_{qp,q^{\prime }p^{\prime }}^{\text{T−ACF}}\left(t,\Delta t\right)=R_{qp,q^{\prime }p^{\prime }}^{\mathrm{LoS}}\left(t,\Delta t\right)+R_{qp,q^{\prime }p^{\prime }}^{\mathrm{ref}}\left(t,\Delta t\right)+R_{qp,q^{\prime }p^{\prime }}^{\mathrm{sca}}\left(t,\Delta t\right) \end{array} \end{array}$$(22)$$\begin{array}{c} \begin{array}{c} R_{qp,q^{\prime }p^{\prime }}^{\text{S−CCF}}\left(d_{T},d_{R},t\right)=R_{qp,q^{\prime }p^{\prime }}^{\mathrm{LoS}}\left(d_{T},d_{R},t\right)+R_{qp,q^{\prime }p^{\prime }}^{\mathrm{ref}}\left(d_{T},d_{R},t\right)+R_{qp,q^{\prime }p^{\prime }}^{\mathrm{sca}}\left(d_{T},d_{R},t\right) \end{array} \end{array}$$

### 3.2. Doppler Power Spectrum Density

The DPSD can be derived from the T-ACF by applying the Fourier transformation in terms of Δt, which can be expressed as follows:(23)$$\begin{array}{c} S_{\mathrm{DPSD},qp,q^{\prime }p^{\prime }}\left(t,f_{D}\right)=\int_{-\infty }^{+\infty }R_{qp,q^{\prime }p^{\prime }}^{T-\mathrm{ACF}}\left(t,\Delta t\right)e^{-j2\pi f_{D}\Delta t}d\Delta t \end{array}$$
where $f_{D}$ denotes the Doppler frequency shift. Similar to Equation (Equation 17), the expression of $S_{\mathrm{DPSD},qp,q^{\prime }p^{\prime }}\left(t,f_{D}\right)$ is given by the following:(24)$$\begin{array}{c} \begin{array}{cc} S_{\mathrm{DPSD},qp,q^{\prime }p^{\prime }}\left(t,f_{D}\right) & =S_{\mathrm{DPSD},qp,q^{\prime }p^{\prime }}^{\mathrm{LoS}}\left(t,f_{D}\right)+S_{\mathrm{DPSD},qp,q^{\prime }p^{\prime }}^{\mathrm{ref}}\left(t,f_{D}\right)+S_{\mathrm{DPSD},qp,q^{\prime }p^{\prime }}^{\mathrm{sca}}\left(t,f_{D}\right)\\ \; & =\int_{-\infty }^{+\infty }R_{qp,q^{\prime }p^{\prime }}^{\mathrm{LoS}}\left(t,\Delta t\right)e^{-j2\pi f_{D}\Delta t}d\Delta t+\int_{-\infty }^{+\infty }R_{qp,q^{\prime }p^{\prime }}^{\mathrm{ref}}\left(t,\Delta t\right)e^{-j2\pi f_{D}\Delta t}d\Delta t\\ \; & +\int_{-\infty }^{+\infty }R_{qp,q^{\prime }p^{\prime }}^{\mathrm{sca}}\left(t,\Delta t\right)e^{-j2\pi f_{D}\Delta t}d\Delta t \end{array} \end{array}$$

## 4. Numerical Results and Analysis

In this section, the statistical properties of the proposed 3D time-varying UAV-MIMO channel model are numerical analyzed in terms of the T-ACF, S-CCF, and DPSD with different channel model parameters and carrier frequencies (e.g., mmWave and THz bands). As there is no relevant measurement data at present, the values of model parameters can only be based on reasonable assumptions. This paper uses the ray-tracing method to verify the correctness of the proposed UAV-based A2A wireless channel model. According to [27,45,46], the parameters for our analysis are set as follows: P=Q= 2, *r* = 5 m, $\alpha_{\mathrm{Tx}}=$180 degrees, $\alpha_{\mathrm{Rx}}=\beta_{\mathrm{Tx}}=\beta_{\mathrm{Rx}}=$0 degrees, $n_{t,\mathrm{ref}}=n_{t,\mathrm{sca}}$ = 2.2 (i.e., the material of the rough surface is concrete), $\sigma_{h,\mathrm{ref}}$ = 0.05 mm, and $\sigma_{h,\mathrm{sca}}$ = 0.15 mm (where $\sigma_{h,\mathrm{sca}}$ is the standard deviation in the height of rough surface in the scattering propagation path). The performance analysis of the THz-UAV wireless communication system in the mmWave and THz bands utilizes carrier frequencies of 60 GHz, 100 GHz, 140 GHz, and 300 GHz, respectively.

Figure 4 depicts the simulation and theoretical results of the T-ACF with different moving speeds of Tx and Rx for the NLoS path (i.e., reflection and scattering paths) and time separations, where $\eta_{\mathrm{ref},qp}=\eta_{\mathrm{ref},q^{\prime }p^{\prime }}$ and $\eta_{\mathrm{sca},qp}=\eta_{\mathrm{sca},q^{\prime }p^{\prime }}$ are 0.6 and 0.4 respectively, and the moving time *t* of Tx and Rx is 0 s, where $H_{0}$ and $D_{0}$ are the initial vertical and horizontal distances between Tx and Rx, respectively. In Figure 4, we can see that the drop rate of T-ACF increases with the increasing of the moving speeds of the Tx and Rx, which is consistent with [8]. The reason is that the phase of CIR changes in direct proportion to the moving speeds of the Tx and Rx. Additionally, the theoretical and simulation results for T-ACF are in close alignment under the same configurations in Figure 4. Then, the correctness of the proposed channel model can be validated.

Figure 5 depicts the theoretical and simulation results of S-CCF for the NLoS path with different $H_{0}$ and element spacings at Tx $d_{T}$ when $\eta_{\mathrm{ref},qp}=\eta_{\mathrm{ref},q^{\prime }p^{\prime }}$ and $\eta_{\mathrm{sca},qp}=\eta_{\mathrm{sca},q^{\prime }p^{\prime }}$ are 0.6 and 0.4 respectively, the moving time *t* of Tx and Rx is 0 s. For this figure, we can see that the S-CCF increases with the increasing of $H_{0}$. In addition, the simulation results of S-CCF are in good agreement with the theoretical results of S-CCF. The correctness of the proposed channel model in this paper can be validated again.

In Figure 6, the theoretical results of the T-ACF for the NLoS path with different carrier frequencies and time separations, including 60 GHz, 200 GHz, 140 GHz, and 300 GHz, are presented, respectively. From this figure, it is shown that the T-ACF decreases with the increasing of the carrier frequency. Moreover, the stationary time separation decreases with the increasing of the carrier frequency, where the stationary time separation is the time separation when the T-ACF is greater than 0.8. The stationary time separation of different carrier frequencies are shown in Table 2.

In Figure 7, it depicts the theoretical results of T-ACF for the NLoS path with different power ratios of reflection and scattering paths and time separations when the moving time *t* of Tx and Rx is 0 s. When the time separation Δt is smaller than 0.075 ms, the T-ACF decreases with the decreasing of the power ratio for the reflection path and the increasing of the power ratio for the scattering path. However, the T-ACF difference of different power ratios for the reflection and scattering paths decreases with the increasing of the time separation Δt.

Figure 8 depicts the theoretical results of S-CCF for the NLoS path with various moving times and different element spacings at Tx $d_{T}$ when $\eta_{\mathrm{ref},qp}=\eta_{\mathrm{ref},q^{\prime }p^{\prime }}$ and $\eta_{\mathrm{sca},qp}=\eta_{\mathrm{sca},q^{\prime }p^{\prime }}$ are 0.6 and 0.4, respectively. In this figure, it shows that the S-CCF increases with the increasing of the moving time for different element spacings of the Tx antenna array when the element spacing of the Tx antenna array $d_{T}$ is smaller than 30λ. It is generally, considered that the S-CCF between two channels is low when the S-CCF value is below 0.6. Table 3 shows that the element spacing of the Tx antenna array for the S-CCF of Tx is below 0.6, with various moving times of Tx and Rx. According to this conclusion of Table 3, the element spacing of Tx antenna array increases with the increasing of the moving time of Tx and Rx when the carrier frequency $f_{c}$, moving speed of Tx $v_{\mathrm{Tx}}$ and Rx $v_{\mathrm{Rx}}$, initial vertical distance $H_{0}$, and horizontal distance $D_{0}$ between Tx and Rx are 300 GHz, 20 m/s, 20 m/s, 50 m, and 100 m, respectively.

According to Equations (19) and (20), the theoretical results of S-CCF for the NLoS path are depicted with various element spacings of Tx and Rx antenna arrays when $\eta_{\mathrm{ref},qp}=\eta_{\mathrm{ref},q^{\prime }p^{\prime }}$ and $\eta_{\mathrm{sca},qp}=\eta_{\mathrm{sca},q^{\prime }p^{\prime }}$ are 0.6 and 0.4 respectively, and the moving time *t* = 0 s, in Figure 9. From this figure, we can see that the S-CCF decreases with the increasing of the element spacing of Tx and Rx antenna arrays. In addition, it is found that the influence of the element spacing at the Rx antenna array on the S-CCF is greater than that of the element spacing at the Tx antenna array on the S-CCF. This is because in the proposed channel model of this paper, most of the scatterer and reflection points are assumed to be distributed around the Rx. In other words, there are few scattering and reflection points around the Tx.

According to Equations (17)–(20) and (22), the theoretical results of the S-CCF for the LoS + NLoS path are presented with different Rician factors *K* and element spacings of the Tx antenna array when $\eta_{\mathrm{ref},qp}=\eta_{\mathrm{ref},q^{\prime }p^{\prime }}$ and $\eta_{\mathrm{sca},qp}=\eta_{\mathrm{sca},q^{\prime }p^{\prime }}$ are 0.6 and 0.4, respectively, and the moving time *t* = 0 s, in Figure 10. As illustrated in this figure, the S-CCF demonstrates an increase with an increase in the Rician factor *K* for various element spacing of Tx antenna array. According to the transmitting theory of the wireless signal [7], the Rician factor *K* can be defined as the ratio between the LoS path power and the NLoS path (i.e., reflection and scattering paths) power. The power from the NLoS component becomes more dominant with the decreasing of the Rician factor *K*; therefore, the value of S-CCF decreases.

According to Equations (23) and (24), the theoretical results of the DPSD are depicted with different carrier frequencies (i.e., 300 GHz, 140 GHz, 60 GHz) and propagation paths (i.e. NLoS path, LoS + NLoS path), when $\eta_{\mathrm{ref},qp}=\eta_{\mathrm{ref},q^{\prime }p^{\prime }}$ and $\eta_{\mathrm{sca},qp}=\eta_{\mathrm{sca},q^{\prime }p^{\prime }}$ are 0.6 and 0.4, respectively, Rician factor K = 5 dB, and the moving time *t* = 0 s, in Figure 11. In this figure, we can observe the following:(1)The DPSDs of LoS + NLoS and NLoS paths exhibited an increase with an increase in the carrier frequency. The result of the NLoS path is more obvious than that of the LoS + NLoS path.(2)The range of DPSDs variations for LoS + NLoS paths is greater than that observed for NLoS paths at different carrier frequencies.

## 5. Conclusions

This paper proposes a 3D time-varying channel model for UAV-based A2A wireless channels in THz band. Due to the random movement of the UAVs at Tx and Rx, the propagation channels are time-varying. Therefore, the time-varying statistical properties of wireless channels are derived and thoroughly analyzed. The proposed model has been validated via the simulation of ray-tracing in terms of the T-ACF and S-CCF, thus confirming the correctness of this channel model. The numerical results demonstrate the following: (1) the T-ACF of the propagation channel decreases with the increasing of the moving speed of Tx and Rx; (2) the S-CCF of the propagation channel increases with the increasing of the vertical distance between Tx and Rx; (3) the stationary time separation decreases with the increasing of the carrier frequency; (4) the element spacing of the Tx antenna array for the low S-CCF increases with the increasing of the moving time of the Tx and Rx. These observations and conclusions of this paper can provide a valuable reference for the UAV-based A2A wireless communication system design in THz band.

## Figures and Tables

**Figure 1 entropy-27-00788-f001:**
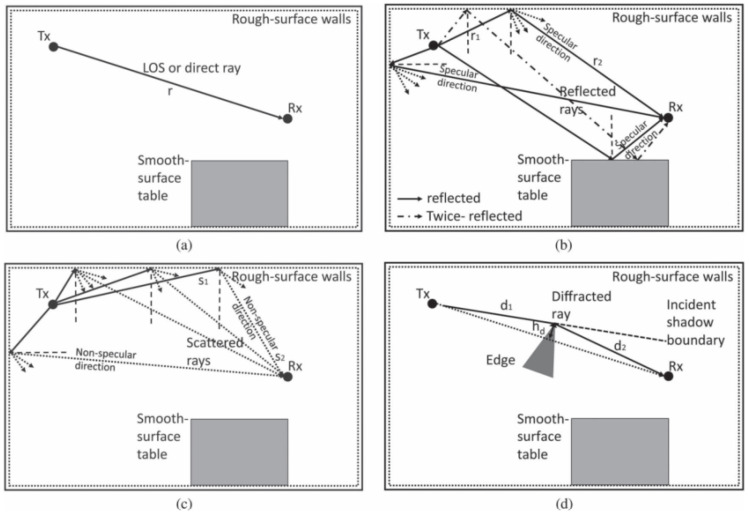
Propagation paths between the transmitter and the receiver: (**a**) LoS path; (**b**) reflection path; (**c**) scattering path; (**d**) diffraction path [30].

**Figure 2 entropy-27-00788-f002:**
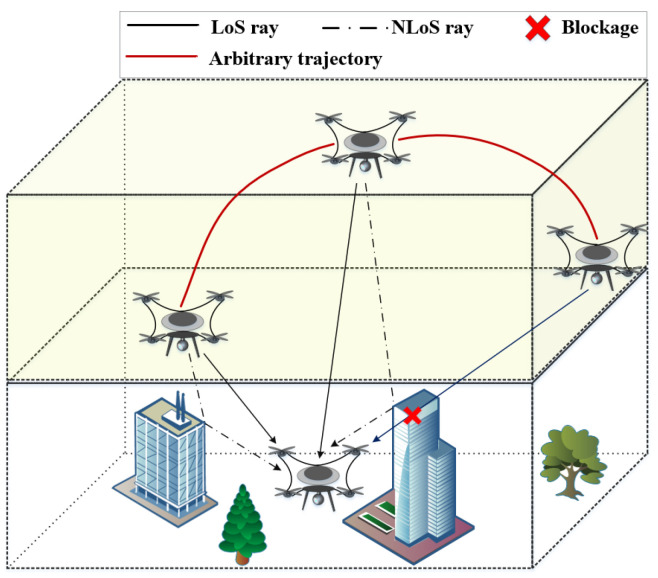
Typical UAV-based THz A2A wireless communications scenario.

**Figure 3 entropy-27-00788-f003:**
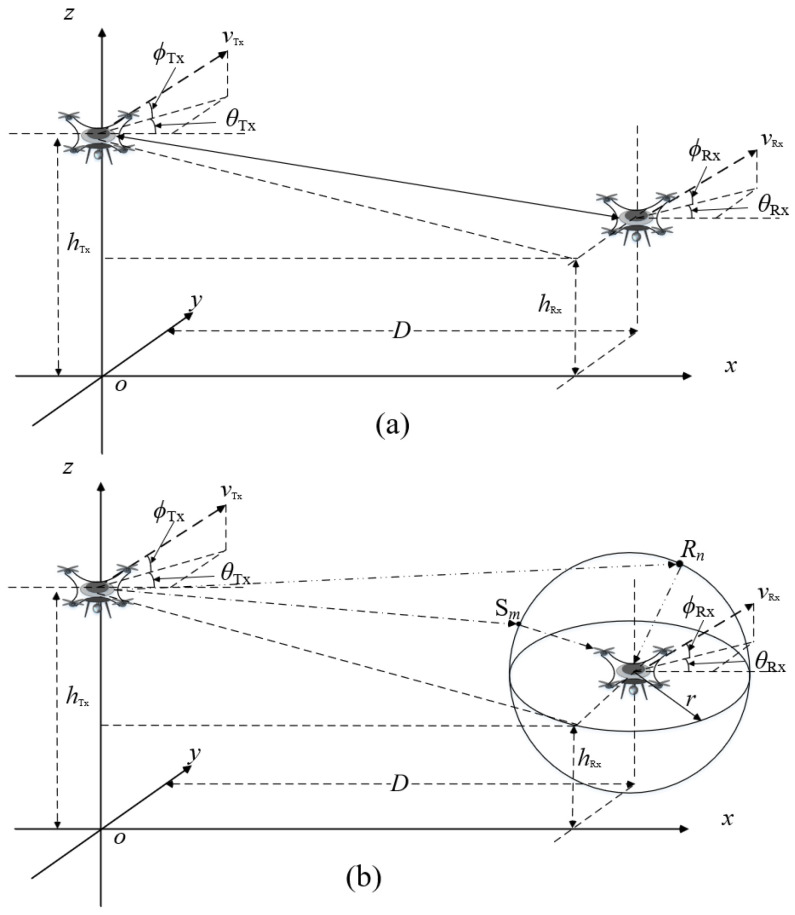
Different propagation paths for the time-varying UAV-based A2A wireless communication system in THz band: (**a**) LoS path; (**b**) NLoS path (i.e., scattering and reflection paths).

**Figure 4 entropy-27-00788-f004:**
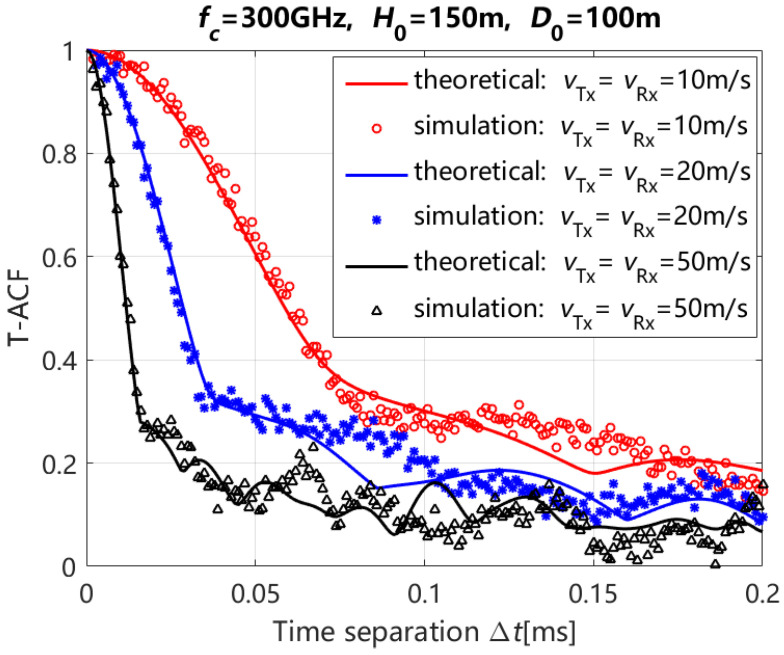
Simulation and theoretical results of the T-ACF for the NLoS path (i.e., reflection and scattering components of NLoS path) with different moving speeds of Tx and Rx.

**Figure 5 entropy-27-00788-f005:**
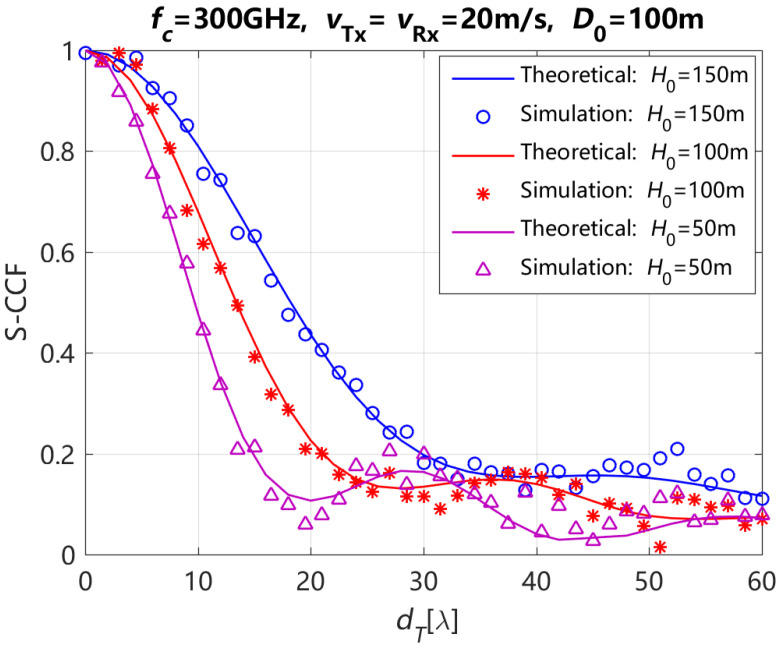
Simulation and theoretical results of the S-CCF for the NLoS path with different initial vertical distances between Tx and Rx $H_{0}$.

**Figure 6 entropy-27-00788-f006:**
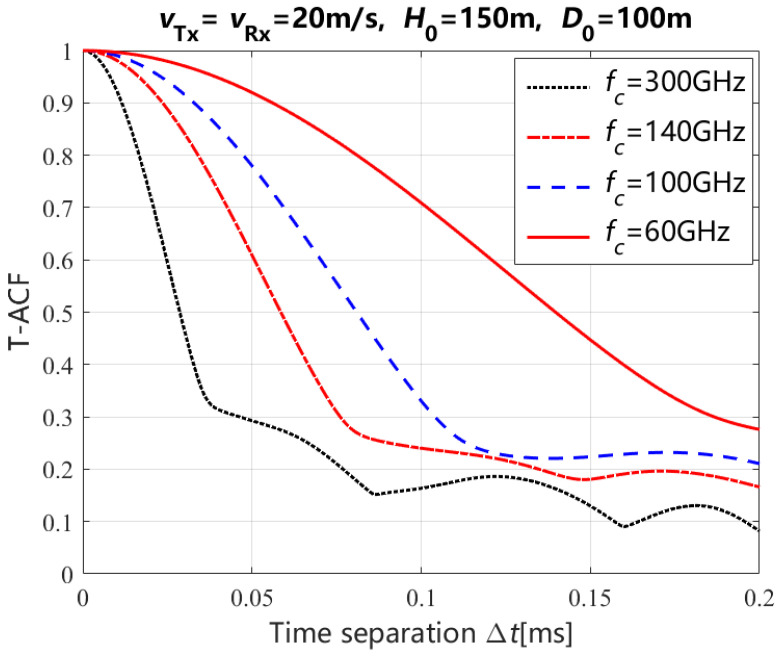
Theoretical results of the T-ACF for the NLoS path with different carrirer frquencies.

**Figure 7 entropy-27-00788-f007:**
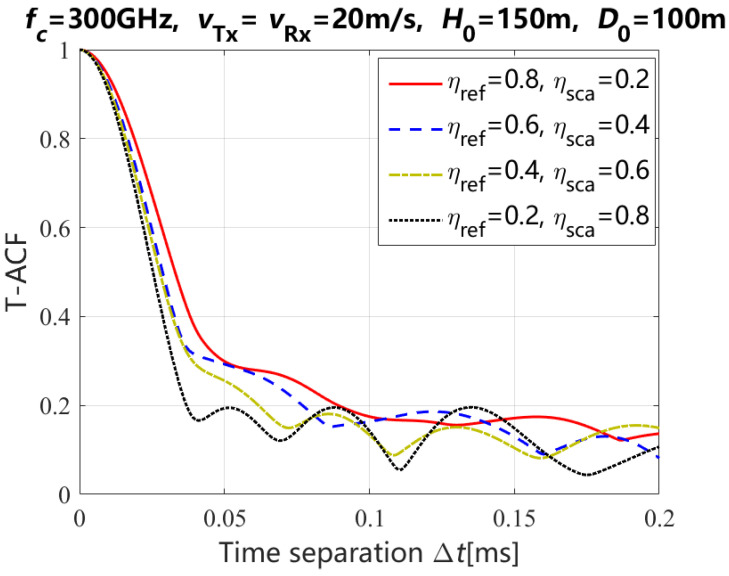
Theoretical results of the T-ACF for the NLoS path with different power ratios of the reflection and scattering paths.

**Figure 8 entropy-27-00788-f008:**
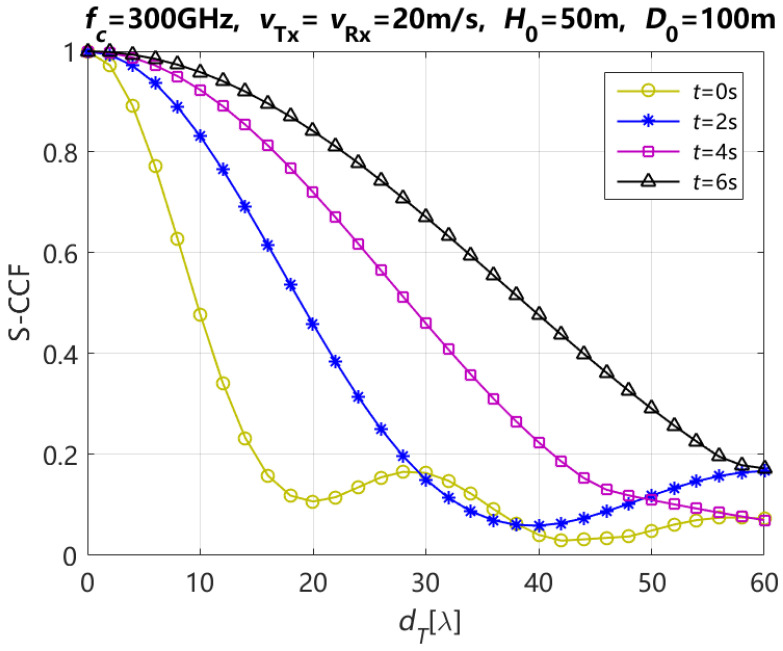
Theoretical results of the S-CCF for the NLoS path with different moving times of Tx and Rx.

**Figure 9 entropy-27-00788-f009:**
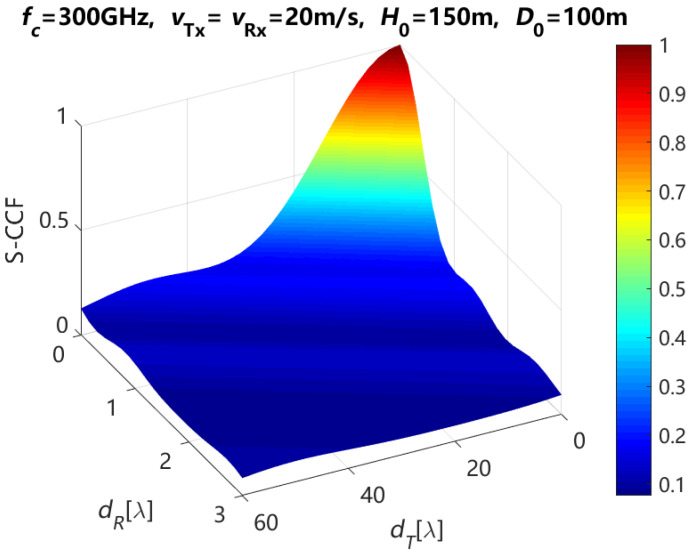
Theoretical results of the S-CCF for the NLoS path with different element spacings of Tx and Rx antenna arrays when the moving time *t* is 0 s.

**Figure 10 entropy-27-00788-f010:**
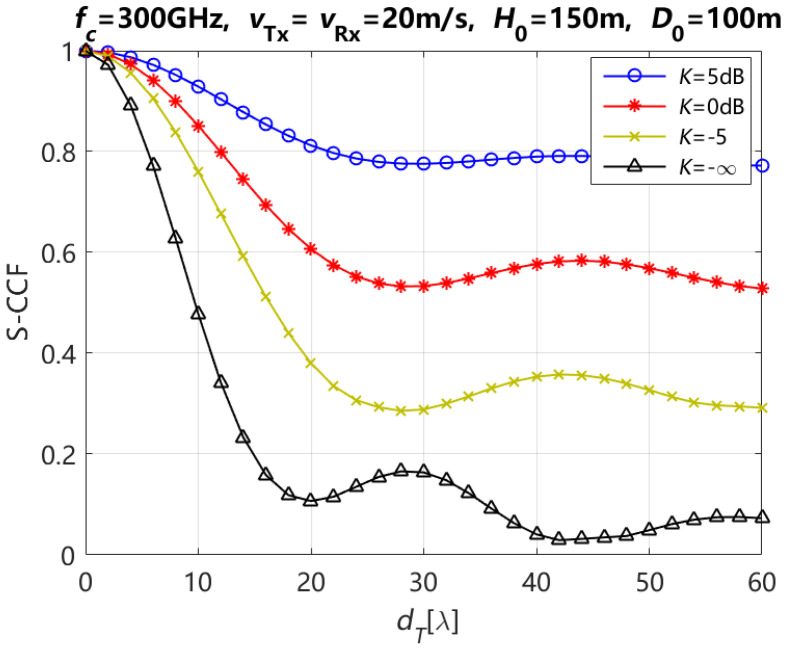
Theoretical results of the S-CCF for the LoS + NLoS path with different Rician factors.

**Figure 11 entropy-27-00788-f011:**
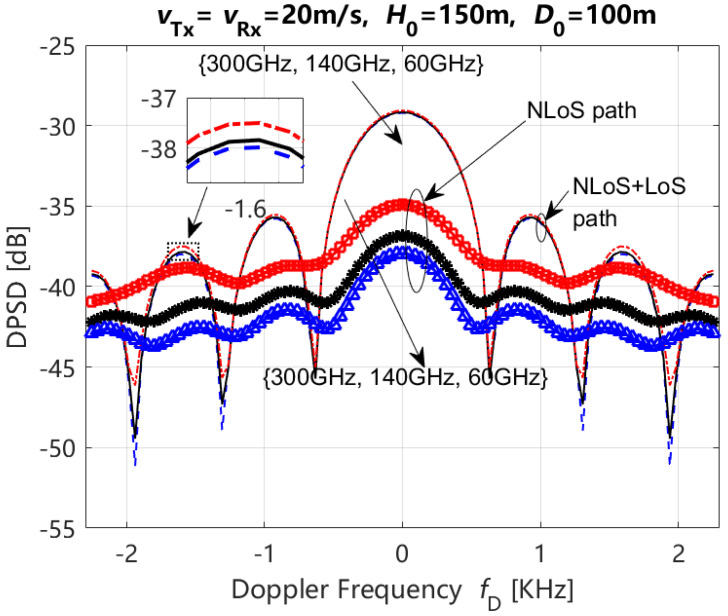
DPSD with different propagation paths (i.e., NLoS path and LoS + NLoS path) and carrier frequencies when the Rician factor is 5 dB.

**Table 1 entropy-27-00788-t001:** Definitions of the parameters for the UAV-Based A2A wireless system.

Symbol	Definition
$\theta_{\mathrm{Tx}}$, $\phi_{\mathrm{Tx}}$	Moving azimuth and elevation angles of Tx
$\theta_{\mathrm{Rx}}$, $\phi_{\mathrm{Rx}}$	Moving azimuth and elevation angles of Rx
$v_{\mathrm{Tx}}$, $v_{\mathrm{Rx}}$	Moving velocities of Tx and Rx
$h_{\mathrm{Tx}}$, $h_{\mathrm{Rx}}$	Vertical heights of Tx and Rx
*D*	Distance at the *x*-axis between Tx and Rx
*r*	Distribution radius of reflectors and scatterers around the Rx
$\theta_{\mathrm{LoS},p}^{\mathrm{AoD}}\left(t\right)$, $\phi_{\mathrm{LoS},p}^{\mathrm{AoD}}\left(t\right)$	Azimuth angle of departure (AAoD) and elevation AoD (EAoD) for the LoS path from the *p*-th antenna element of Tx to the *q*-th antenna element of Rx
$\theta_{\mathrm{LoS},q}^{\mathrm{AoA}}\left(t\right)$, $\phi_{\mathrm{LoS},q}^{\mathrm{AoA}}\left(t\right)$	Azimuth angle of arrival (AAoA) and elevation AoA (EAoA) for the LoS path from the *p*-th antenna element of Tx to the *q*-th antenna element of Rx
$\theta_{\mathrm{ref},n,p}^{\mathrm{AoD}}\left(t\right)$, $\phi_{\mathrm{ref},n,p}^{\mathrm{AoD}}\left(t\right)$	AAoD and EAoD for the reflection path from the *p*-th antenna element of Tx to *n*-th reflector, then to the *q*-th antenna element of Rx
$\theta_{\mathrm{ref},n,q}^{\mathrm{AoA}}\left(t\right)$, $\phi_{\mathrm{ref},n,q}^{\mathrm{AoA}}\left(t\right)$	AAoA and EAoA for the reflection path from the *p*-th antenna element of Tx to *n*-th reflector, then to the *q*-th antenna element of Rx
$\theta_{\mathrm{sca},m,p}^{\mathrm{AoD}}\left(t\right)$, $\phi_{\mathrm{sca},m,p}^{\mathrm{AoD}}\left(t\right)$	AAoD and EAoD for the scattering path from the *p*-th antenna element of Tx to *m*-th scatterer, then to the *q*-th antenna element of Rx
$\theta_{\mathrm{sca},m,q}^{\mathrm{AoA}}\left(t\right)$, $\phi_{\mathrm{sca},m,q}^{\mathrm{AoA}}\left(t\right)$	AAoA and EAoA for the scattering path from the *p*-th antenna element of Tx to *m*-th scatterer, then to the *q*-th antenna element of Rx
$\vartheta_{\mathrm{sca},m}$, $\vartheta_{\mathrm{ref},n}$	Random phases of scattering and reflection propagations caused by $S_{m}$ and $R_{n}$, and they can be assumed to independently, uniformly, and randomly distributed in the interval of (−π,π)

**Table 2 entropy-27-00788-t002:** Channel stationary time separation with different carrier frequencies.

Carrier Frequency $f_{c}$	Stationary Time Separation Δt
300 GHz	0.016 ms
140 GHz	0.034 ms
100 GHz	0.047 ms
60 GHz	0.081 ms

**Table 3 entropy-27-00788-t003:** Element spacing of Tx antenna array with low S-CCF for different moving times of Tx and Rx.

Moving Time *t*	Element Spacing of Tx $d_{T}$
0 s	8λ
2 s	16λ
4 s	24λ
6 s	34λ

## Data Availability

Data is contained within the article.

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
