# Peer review of "Three-Dimensional Non-Stationary MIMO Channel Modeling for UAV-Based Terahertz Wireless Communication Systems"

_entropy, 2025, doi:10.3390/e27080788_

Round 1
Reviewer 1 Report
Comments and Suggestions for Authors
See the attached file.

See the attached file.
Author Response
Thank you for your comments concerning our manuscript entitled “Three-Dimensional Non-Stationary MIMO Channel Modeling for UAV-Based Terahertz Wireless Communication Systems” (Manuscript ID: entropy-33676893). Those comments are all valuable and very helpful for revising and improving our paper, as well as the important guiding significance to our researches. We have studied your comments carefully and have made correction which we hope meet with your approval. The main corrections in the paper and the responds to your comments are at the file of "author-coverletter-47230655.v1.pdf".

Reviewer 2 Report
Comments and Suggestions for Authors
Please provide a response to the reviewer's comment in the report. The response should include both an explanation and the corresponding action taken, rather than simply directing the reviewer to the revised manuscript.
- Lacks real-world measurements. Validation relies solely on simulations (ray-tracing), limiting empirical credibility.
- Assumes single-bounce NLoS paths only, ignoring multi-hop reflections/scattering – may oversimplify complex THz propagation.
-
Model parameters (e.g., scatterer distribution, material properties) are based on assumptions due to scarce THz data, potentially affecting realism.
-
Simulations focus on ≤300GHz, but THz (>300GHz) channel behaviors remain uncertain without higher-frequency validation.
-
Asserts scatterers/reflectors exist only near Rx (Page 5), yet UAVs at both ends are mobile – this ignores potential scattering around Tx, especially at THz frequencies with omnidirectional small-scale roughness.
-
Ray-tracing validation cites Refs. [46-48], but these focus on sub-300GHz scenarios. No justification for extending to THz (>300GHz) bands where material properties differ significantly.
-
Ref. [28]: Identical to Ref. [25] (both cite Rappaport et al. 2019) – likely duplication.
-
Table 3: Recommends Tx antenna spacing up to 34λ at t=6s (e.g., 34cm@300GHz). Fails to consider:
-
Physical feasibility: 34λ spacing requires impractical UAV array size (e.g., >1m at 300GHz).
-
Frequency dependence: Spacing in "wavelengths (λ)" ignores absolute frequency impact on spatial correlation.
-
-
Coherent summation (Eqn. 2) contradicts THz channel physics in [30][39].
Doppler unit error (Eqn. 4) violates fundamental EM theory. -
Validates model via ray-tracing (Section 4) but uses mmWave parameters (Ref. [46-48]) for THz bands.
-
Flaw: Material properties (e.g., roughness, permittivity) differ significantly at THz frequencies.
-
Impact: Validation results (Fig. 4-5) may be invalid for >100GHz scenarios.
-
Author Response
Thank you for your comments concerning our manuscript entitled “Three-Dimensional Non-Stationary MIMO Channel Modeling for UAV-Based Terahertz Wireless Communication Systems” (Manuscript ID: entropy-33676893). Those comments are all valuable and very helpful for revising and improving our paper, as well as the important guiding significance to our researches. We have studied your comments carefully and have made correction which we hope meet with your approval. The main corrections in the paper and the responds to your comments are at the files "author-coverletter-48025669.v1.pdf".

Round 2
Reviewer 2 Report
Comments and Suggestions for Authors
It is good enough to publish the Entropy.